# Increased mask adherence after important politician infected with COVID-19

Deborah A. Cohen[1]*, Meghan Talarowski[2], Olaitan Awomolo[2], Bing Han[3], Stephanie Williamson[3], Thomas L. McKenzie[4]

1 Kaiser Permanente Southern California Research and Evaluation, Pasadena, CA, United States of America, 2 Studio Ludo, Philadelphia, PA, United States of America, 3 RAND Corporation, Santa Monica, CA, United States of America, 4 Emeritus, San Diego State University, San Diego, CA, United States of America

* Deborah.a.cohen@kp.org

## Abstract

### Objectives

To quantify changes in adherence to mask and distancing guidelines in outdoor settings in Philadelphia, PA before and after President Trump announced he was infected with COVID-19.

### Methods

We used Systematic Observation of Masking Adherence and Distancing (SOMAD) to assess mask adherence in parks, playgrounds, and commercial streets in the 10 City Council districts in Philadelphia PA. We compared adherence rates between August and September 2020 and after October 2, 2020.

### Results

Disparities in mask adherence existed by age group, gender, and race/ethnicity, with females wearing masks correctly more often than males, seniors having higher mask use than other age groups, and Asians having higher adherence than other race/ethnicities. Correct mask use did not increase after the City released additional mask guidance in September but did after Oct 2. Incorrect mask use also decreased, but the percentage not having masks at all was unchanged.

### Conclusions

Vulnerability of leadership appears to influence population behavior. Public health departments likely need more resources to effectively and persuasively communicate critical safety messages related to COVID-19 transmission.

**Data Availability Statement:** The data will be made accessible on this website: https://www.kp-scalresearch.org/somad/ when accepted for publication.

**Funding:** This study was supported in part by NHLBI # R01HL145145 (PI is DC). The study sponsor played no role in the study design; in the collection, analysis, and interpretation of data; in the writing of the report; and in the decision to submit the article for publication. National Heart Lung Blood Institute https://www.nhlbi.nih.gov/.

**Competing interests:** The authors have declared that no competing interests exist.

## Introduction

Prior to widespread vaccine availability, the only way to prevent the spread of COVID-19 was by wearing a mask, maintaining a physical distance of at least six feet from others, and frequent handwashing. Multiple modeling studies of the spread of COVID-19 support the importance of wearing masks and maintaining a physical distance from others [1–4]. One modeling study suggested that 80% compliance with mask wearing would reduce mortality from COVID-19 by up to 45% [2], and it has been suggested that masks may reduce the size of the inoculum, leading to milder infections [5].

The science demonstrating the effectiveness of masking is very strong [6], yet this protective behavior has become politicized. Some consider mandates to wear masks a violation of individual freedom. In spite of the persistent spread of the infection and an increasing death tally in countries like Brazil and the United States, many do not wear masks in public settings. In countries like South Korea, Japan and China, where adherence to masking mandates were high, the case rates were considerably lower than in the US and Brazil [7].

Understanding adherence to mask and distancing guidelines may be critical to controlling disease spread in the absence of a vaccine or if a large proposal of the population refuses to accept vaccination. Most studies on adherence rely on either modeling [1–4], documenting the presence or absence of policies [8–10], or self-report [11]. Direct observation has repeatedly been demonstrated to be a reliable method of measuring a variety of individual characteristics and behaviors, including the intensity of physical activity and human interactions [12–14]. The technique entails data collectors recording a limited number of visible characteristics of the individuals they observe. Respondent burden and reactivity are both eliminated as observers do not interact with subjects. When conducted in public settings, systematic observations studies are generally categorized as exempt by human subjects' protection committees.

Given the controversy about mask use, we wondered whether President Trump's COVID-19 infection might influence adherence to recommendations to wear masks in public settings. We capitalized on our ongoing surveillance of mask wearing in Philadelphia, the city where most of our staff are located, to determine whether adherence changed after the President reported his disease state. Understanding which factors promote better adherence to masking and distancing guidelines is critical for controlling virus spread.

## Methods

We employed Systematic Observation of Mask Adherence and Distancing (SOMAD), a direct observation tool to document the number of people wearing masks correctly and keeping at least six feet away from others. The reliability of SOMAD was assessed to have less than 10% measurement errors for each variable between two independent observers and less than 1.2% when aggregated by day [14]. (The tool is available on https://www.kp-scalresearch.org/somad/).

Observations were conducted in parks, playgrounds, and commercial streets in each of Philadelphia's 10 City Council districts, a total of 30 sites. Locations were chosen based on their having a high number of people passing through the areas under observation. Each site was observed for one hour on both a weekday and a weekend day, with each observed at the same time of day on each occasion for a total of 60 observation hours in August and another 60 hours between September 23 and October 11, 2020. Trained data collectors observed individuals in these settings, recording their characteristics and behaviors including: age group (infant/toddler ages 0–2), child (3–12), teen (13–19), adult (20–59), and senior (> = 60); gender; apparent race/ethnicity (white, black, Asian, Latino, undetermined), and mask adherence (correct use; incorrect use, but mask visible; no mask). Correct mask use was defined as having the

mask covering both mouth and nose. Incorrect use was defined as either mouth or nose exposed. We also collected information simultaneously on each person's physical activity level (sedentary, moderate, and vigorous), mode of transport (on wheels or not), group size (alone, 2, 3–5, 6–9, 10+), and physical distancing (>6 feet from others or not). At each location observers noted whether there was crowding, defined as having more people than would make it possible to stay at least 6 feet apart from others. All data were entered using a Google form. Given observers did not interact with human subjects, the study was deemed exempt by the RAND IRB.

Observations took place between August 11 and August 30, 2020 and between September 23 and October 11, 2020. We compared mask adherence in August and September and after October 2, 2020, the date the President's COVID-19 infection was made public. Our analysis includes descriptive statistics as well as a Generalized Estimating Equations (GEE) model controlling for all the eight individual variables observed, as well as the setting, population density, percentage of households in poverty in the council district, and the time of the observation.

## Results

During August 2020 we observed 4606 individuals across the 30 locations. Overall, 43.2% wore it correctly, 16.7% wore it incorrectly, and 40.2% did not wear a mask at all. (See Table 1). Patterns of disparities in correct mask use persisted over time. From August through the beginning of October females had higher correct mask usage than males (58.7% vs. 45.3%, p < .0001); among the four age groups seniors had the highest correct use (57.8%) while teens had only 37.5% correct use (p < .0001); Asians had the highest adherence among racial/ethnic groups (60.3%) and Hispanic/Latinex the lowest (38.2%) (P < .0001).

Between September 23 and October 1, of 2641 people observed, 36.7% did not wear a mask, 44.1% wore it correctly, and 19.2% wore it incorrectly, a non-significant change from August, 2020 (p = .31). However, from October 2 through October 11, of 2473 observed, correct mask use increased to 51.4% while incorrect use dropped to 9.7% (p < .0001). Correct mask use was observed among males (17%), females (16%), younger adults (15%), seniors (18%), whites (24%), and those categorized as Latinx (53%). (See Table 1).

After controlling for individual characteristics, time and setting variables, multiple differences in mask adherence were seen (Table 2). Across age groups, senior used masks correctly the most. Females used them more than males and Asians wore masks correctly more often than all other racial/ethnic groups. Those engaged in moderate physical activity wore masks correctly more often than those who were sedentary or in vigorous activity. Consistent with this, those on wheels (e.g. bicycles, roller blades, strollers) used masks less often than those not on wheels. No differences in mask use were seen based on group size, weekdays vs. weekend days, percentage poverty level of the neighborhood setting being observed, or whether people kept at least a 6-foot distance from others. Neighborhood population density, however, was positively associated with higher correct mask use. Those observed on commercial streets were more likely to wear masks correctly compared to those in parks or playgrounds. Our model confirmed mask adherence was significantly higher in October after the President's infection was announced than in both August and September (adjusted odds ratio = 1.377, p = .0097).

## Discussion

The City of Philadelphia Health Dept engaged in extraordinary efforts to promote mask use throughout the summer and issued additional detailed instructions on appropriate wear on September 15, 2020. In spite of these efforts, increased correct mask use was not seen until after the President's infection was announced on October 2, 2020.

**Table 1.  Mask use adherence before and after Oct 2, 2020, Philadelphia PA.**

| | Before 02 October 2020 | | | | | 02 October 2020 and after | | | | | |
| | N | Overall | Mask Correct | Mask Incorrect Visible | No Mask Seen | N | Overall | Mask Correct | Mask Incorrect Visible | No Mask Seen | P value comparing before/after Oct 2 |
|---|---|---|---|---|---|---|---|---|---|---|---|
| (N) | 2641 | | 1166 | 507 | 968 | 2473 | | 1272 | 239 | 962 | |
| Overall Mask use | | | 44.1% | 19.2% | 36.7% | | | 51.4% | 9.7% | 38.9% | < .0001 |
| **Gender** | | | | | | | | | | | |
| Male | 1399 | 54.1% | 38.7% | 19.4% | 41.9% | 1325 | 53.6% | 45.3% | 9.8% | 44.9% | < .0001 |
| Female | 1168 | 45.2% | 50.6% | 18.1% | 31.3% | 1144 | 46.3% | 58.7% | 9.5% | 31.7% | < .0001 |
| Non-Binary/Unknown | 18 | 0.7% | 11.1% | 33.3% | 55.6% | 4 | 0.2% | 0.0% | 0.0% | 100.0% | 0.2474 |
| **Age Group** | | | | | | | | | | | |
| Toddler | 93 | 3.5% | 7.5% | 2.2% | 90.3% | 59 | 2.4% | 11.9% | 0.0% | 88.1% | 0.3632 |
| Child | 271 | 10.3% | 34.3% | 9.6% | 56.1% | 233 | 9.5% | 42.1% | 5.6% | 52.4% | 0.0858 |
| Teen | 100 | 3.8% | 42.0% | 15.0% | 43.0% | 120 | 4.9% | 37.5% | 12.5% | 50.0% | 0.5769 |
| Adult | 1885 | 71.7% | 46.9% | 20.9% | 32.1% | 1791 | 72.8% | 54.1% | 9.9% | 36.0% | < .0001 |
| Senior | 279 | 10.6% | 49.1% | 22.6% | 28.3% | 256 | 10.4% | 57.8% | 11.3% | 30.9% | 0.0025 |
| **Race/ethnicity** | | | | | | | | | | | |
| Non-Hispanic White | 1419 | 55.1% | 46.4% | 10.9% | 42.6% | 1422 | 57.6% | 56.7% | 6.5% | 36.8% | < .0001 |
| Non-Hispanic Black/ African American | 771 | 29.9% | 40.7% | 31.3% | 28.0% | 812 | 32.9% | 42.5% | 14.5% | 43.0% | < .0001 |
| Non-Hispanic Asian | 151 | 5.9% | 64.9% | 19.2% | 15.9% | 121 | 4.9% | 60.3% | 5.8% | 33.9% | < .0001 |
| Hispanic/Latinx | 221 | 8.6% | 24.9% | 24.9% | 50.2% | 110 | 4.5% | 38.2% | 18.2% | 43.6% | 0.0378 |
| Unknown/unable to determine | 15 | 0.6% | 33.3% | 40.0% | 26.7% | 5 | 0.2% | 60.0% | 20.0% | 20.0% | 0.5594 |
| **Activity level** | | | | | | | | | | | |
| Sedentary | 218 | 8.3% | 26.6% | 28.4% | 45.0% | 183 | 7.4% | 23.5% | 9.8% | 66.7% | < .0001 |
| Moderate | 2282 | 86.4% | 47.9% | 17.5% | 34.6% | 2184 | 88.3% | 54.7% | 9.8% | 35.5% | < .0001 |
| Vigorous | 141 | 5.3% | 11.3% | 31.9% | 56.7% | 106 | 4.3% | 33.0% | 6.6% | 60.4% | < .0001 |
| **Transportation mode** | | | | | | | | | | | |
| On wheels | 210 | 8.3% | 23.8% | 10.0% | 66.2% | 194 | 7.9% | 29.4% | 3.6% | 67.0% | 0.0282 |
| Not on wheels | 2327 | 91.7% | 47.2% | 17.4% | 35.5% | 2268 | 92.1% | 53.4% | 10.2% | 36.4% | < .0001 |
| **Group size** | | | | | | | | | | | |
| Not in a group | 1180 | 45.6% | 43.7% | 21.9% | 34.3% | 1111 | 45.0% | 54.0% | 10.4% | 35.6% | < .0001 |
| group of 2 | 673 | 26.0% | 50.5% | 18.1% | 31.4% | 799 | 32.3% | 51.4% | 8.8% | 39.8% | < .0001 |
| group of 3 to 5 | 555 | 21.5% | 43.4% | 11.9% | 44.7% | 517 | 20.9% | 49.1% | 8.9% | 42.0% | 0.0984 |
| group of 6 to 9 | 135 | 5.2% | 22.2% | 20.0% | 57.8% | 25 | 1.0% | 20.0% | 24.0% | 56.0% | 0.8946 |
| group of 10 or more | 43 | 1.7% | 11.6% | 44.2% | 44.2% | 19 | 0.8% | 0.0% | 5.3% | 94.7% | 0.0009 |
| **Keep > 6ft distance from others** | | | | | | | | | | | |
| Yes | 1224 | 46.8% | 44.0% | 21.2% | 34.8% | 1202 | 48.8% | 51.7% | 10.1% | 38.1% | < .0001 |
| No | 1394 | 53.3% | 44.8% | 16.6% | 38.7% | 1259 | 51.2% | 51.0% | 9.1% | 40.0% | < .0001 |
| **Setting** | | | | | | | | | | | |
| Commercial Street | 1176 | 44.5% | 48.2% | 27.0% | 24.7% | 1266 | 51.2% | 60.8% | 11.1% | 28.0% | < .0001 |
| Neighborhood Park | 925 | 35.0% | 40.3% | 15.0% | 44.6% | 831 | 33.6% | 49.3% | 8.4% | 42.2% | < .0001 |
| Playground | 540 | 20.5% | 41.9% | 9.3% | 48.9% | 376 | 15.2% | 24.5% | 7.4% | 68.1% | < .0001 |

Although this is a serial cross-sectional observational study and the same people were not observed on each occasion, the increase in correct mask wearing appears to be among those who already had masks, because there was virtually no change in the proportion of those without a mask. It's possible that the news may have instilled increased fear of the disease, resulting

**Table 2. Model of mask use over time.**

| Variables | estimate | SD | 95% C.I. | | p-value |
|---|---|---|---|---|---|
| Intercept | -2.80 | -0.77 | -4.83 | -0.77 | 0.007 |
| Toddler | -2.21 | -1.67 | -2.76 | -1.67 | < .0001 |
| Child | -0.64 | -0.27 | -1.01 | -0.27 | 0.0007 |
| Teen | -0.91 | -0.54 | -1.27 | -0.54 | < .0001 |
| Adult | -0.40 | 0.11 | -0.62 | -0.18 | 0.0004 |
| Senior | ref | – | – | – | – |
| Female | 0.49 | 0.06 | 0.37 | 0.61 | < .0001 |
| Non-Binary/Unknown | -0.19 | 0.46 | -1.10 | 0.72 | 0.69 |
| Male | ref | – | – | – | – |
| Non-Hispanic Black/African American | -0.42 | 0.17 | -0.74 | -0.09 | 0.01 |
| Non-Hispanic Asian | 0.52 | 0.19 | 0.16 | 0.89 | 0.005 |
| Hispanic/Latinx | -0.74 | 0.22 | -1.18 | -0.31 | 0.0009 |
| Unknown/unable to determine | 0.23 | 0.15 | -0.07 | 0.53 | 0.13 |
| Non-Hispanic White | ref | – | – | – | – |
| Sedentary | -0.19 | 0.26 | -0.70 | 0.31 | 0.46 |
| Moderate | 0.83 | 0.22 | 0.40 | 1.26 | 0.0002 |
| Vigorous | ref | – | – | – | – |
| Not in a group | 0.84 | 0.99 | -1.10 | 2.78 | 0.40 |
| group of 2 | 0.88 | 0.95 | -0.99 | 2.74 | 0.36 |
| group of 3 to 5 | 0.92 | 0.99 | -1.02 | 2.85 | 0.35 |
| group of 6 to 9 | 0.31 | 1.01 | -1.68 | 2.29 | 0.76 |
| group of 10 or more | ref | – | – | – | – |
| Physically distanced | -0.06 | 0.10 | -0.26 | 0.14 | 0.55 |
| Not physically distanced | ref | – | – | – | – |
| On wheels | -0.78 | 0.23 | -1.23 | -0.33 | 0.0007 |
| Not on wheels | ref | – | – | – | – |
| Weekend | -0.17 | 0.12 | -0.40 | 0.06 | 0.14 |
| Weekday | ref | – | – | – | – |
| % households below poverty | 0.003 | 0.01 | -0.02 | 0.02 | 0.79 |
| Commercial Street | 1.05 | 0.27 | 0.52 | 1.58 | 0.0001 |
| Neighborhood Park | 0.81 | 0.29 | 0.25 | 1.38 | 0.0050 |
| Playground | ref | – | – | – | – |
| Population density | 0.05 | 0.02 | 0.01 | 0.09 | 0.0250 |
| Prior to October 2, 2020 | 0.11 | 0.11 | -0.10 | 0.32 | 0.3057 |
| On or after Oct 2, 2020 | 0.35 | 0.14 | 0.09 | 0.62 | 0.0097 |
| August, 2020 | ref | – | – | – | – |

in those having masks being more careful in their appropriate use (e.g., both nose and mouth covered) in public settings. The rise in correct mask use after the President's COVID-19 infection suggests that the behavior of our leaders has a significant impact on population adherence to public health guidelines.

Although the City of Philadelphia did issue guidance about mask adherence, it is likely that this did not receive as much attention as the President's infections which made headlines in the national news for many days. It's possible that the prominence of the news was an even more likely trigger for increased adherence.

Considering the contagiousness and virulence of COVID-19, the continued lack of mask use among 36% of those observed is concerning. Although outdoor settings are considered

lower risk than indoor settings, the spaces observed were all public outdoor areas where people could come into contact with others and be exposed to aerosolized droplets. Even though outdoor settings provide better ventilation when one is not distanced or protected by masks, an increasing amount of time spent in close proximity to others also increases the risk of transmission, even in an outdoor setting [15].

Because the risk of transmission is a function of both dosage and duration of exposure, settings where people spend time, like in parks or playgrounds are places where masks should be worn. Yet people were less likely to wear them in parks than on commercial streets, possibly because they may have more control over distancing in these settings.

Meanwhile, important and yet unanswered questions include whether mask wearing in one setting is a good proxy for mask adherence in other areas and whether mask adherence in outdoor settings is its own predictor of transmission risk. Further, it is important to determine whether seeing others without masks establishes a norm or signals that mask adherence is unimportant, factors that could potentially undermine COVID-19 control efforts.

The study has several limitations. All the data are based on observations and estimates from trained field staff. Although the methods have high reliability, there may have been some misclassifications. In outdoor settings the risk for transmission of COVID-19 is lower than indoors, so mask adherence in these settings may not predict transmission. We could not know the relationship of people who were not wearing masks and were not distanced from each other. It is possible they lived in the same household and thus the guidelines were not applicable to their situation. This is also an observational study, not a randomized controlled trial, so causal inferences are speculative.

Our sample size was based on prior studies using direct observations, where anywhere between 6 to 50 sites (e.g., neighborhood parks, recreation centers), have been selected for direct observations. Each observation hour was expected to allow documentation of at least 60 individuals. We expected that we would need to observe at least 1000 individuals, and this did turn out to be sufficient. The number of locations and sample size was also influenced by the limited manpower available.

Although we observed increased adherence after President Trump was infected and not after the City Health Dept issued additional mask adherence guidance, we can only hypothesize that the prominence of the news and the real-life example showing how non-adherence leads to infection is what inspired this change. Certainly, publicity and widespread dissemination of information and guidance has been shown to be a critical predictor of behavior change in many other public health interventions [16–19]. Personalizing information is also an effective advertising technique as influencers and testimonials are known to be powerful methods for promoting behavior change [20].

There are multiple implications of our findings. The relatively low adherence rates in commercial settings provided a strong rationale for the Dept. of Public Health to act, which they did. However, their resources did not match the avalanche of publicity that accompanied the news of President Trump's infection. This suggests that local public health agencies need more resources for information campaigns and enforcement activities. Given the multiple sources of misinformation about the pandemic, directing resources to disseminate clear and factual information about prevention is sorely needed. Given that the percentage of persons with no masks remained consistently high even after the President became infected, suggests that even widely disseminated information campaigns may be insufficient to obtain compliance.

The need for mask wearing is likely to continue, not only due to variants of COVID-19, like Delta, but also due to the potential emergence of other viruses, given our recent experience with H1NI and MERS just in the past few decades [21,22]. Additional consideration should be

made for increasing monitoring and possibly stronger enforcement efforts in higher risk public settings where people may spend an extended time in close proximity to others.

## Author Contributions

**Conceptualization:** Deborah A. Cohen, Meghan Talarowski, Thomas L. McKenzie.

**Data curation:** Olaitan Awomolo, Bing Han, Stephanie Williamson.

**Formal analysis:** Bing Han, Stephanie Williamson.

**Funding acquisition:** Deborah A. Cohen.

**Investigation:** Deborah A. Cohen, Olaitan Awomolo, Bing Han.

**Methodology:** Deborah A. Cohen, Meghan Talarowski, Olaitan Awomolo, Bing Han, Thomas L. McKenzie.

**Project administration:** Deborah A. Cohen, Meghan Talarowski, Olaitan Awomolo.

**Resources:** Meghan Talarowski.

**Software:** Olaitan Awomolo.

**Supervision:** Deborah A. Cohen, Meghan Talarowski, Olaitan Awomolo.

**Validation:** Deborah A. Cohen, Olaitan Awomolo, Bing Han, Stephanie Williamson.

**Writing – original draft:** Deborah A. Cohen.

**Writing – review & editing:** Meghan Talarowski, Olaitan Awomolo, Bing Han, Stephanie Williamson, Thomas L. McKenzie.

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
