## [Decision Letter · Decision Letter 0]

7 Jul 2021

PONE-D-20-38624

Increased Mask Adherence after President Trump Infected with COVID-19

PLOS ONE

Dear Dr. Cohen,

Thank you for submitting your manuscript to PLOS ONE. After careful consideration, we feel that it has merit but does not fully meet PLOS ONE’s publication criteria as it currently stands. Therefore, we invite you to submit a revised version of the manuscript that addresses the points raised during the review process.

Your manuscript has undergone the peer-review process and the reviewers have provided their comments/suggestions. Kindly address these points/concerns before we make a decision.

We look forward to receiving your revised manuscript.

Kind regards,

Kingston Rajiah

Academic Editor

PLOS ONE

Journal Requirements:

2. Thank you for stating the following in the Competing Interests/Financial Disclosure section: 

No (Competing Interests) / This study was supported in part by NHLBI # R01HL145145  (PI  is DC). The study sponsor played no role in the study design; in the collection, analysis, and interpretation of data; in the writing of the report; and in the decision to submit the article for publication.  

National Heart Lung Blood Institute  https://www.nhlbi.nih.gov/. (Financial Disclosure)

We note that one or more of the authors are employed by a commercial company: RAND Corporation. 

This study was supported in part by NHLBI # R01HL145145.

This study was supported in part by NHLBI # R01HL145145.

 NO

8.We note you have included a table to which you do not refer in the text of your manuscript. Please ensure that you refer to Table 1 in your text; if accepted, production will need this reference to link the reader to the Table.

Reviewers' comments:

Reviewer's Responses to Questions

**Comments to the Author**

1. Is the manuscript technically sound, and do the data support the conclusions?

Reviewer #1: Yes

Reviewer #2: Partly

Reviewer #3: Partly

2. Has the statistical analysis been performed appropriately and rigorously? 

Reviewer #1: Yes

Reviewer #2: I Don't Know

Reviewer #3: Yes

3. Have the authors made all data underlying the findings in their manuscript fully available?

Reviewer #1: Yes

Reviewer #2: Yes

Reviewer #3: Yes

4. Is the manuscript presented in an intelligible fashion and written in standard English?

Reviewer #1: Yes

Reviewer #2: Yes

Reviewer #3: Yes

5. Review Comments to the Author

Reviewer #1: The article presents a relevant and innovative theme. Some small adjustments must be made, they are:

- Improve the introduction, putting an international panorama on the theme (how is the relation of the use of masks in other countries)?

- Improve at the end of the discussion, what are the limitations of the study.

- Create a paragraph at the end of the discussion with the practical / clinical implications of your study.

- Improve completion by detailing in a topic.

Reviewer #2: I am pleased to share my comment, for the article entitled: Increased Mask Adherence after President Trump Infected with COVID-19

Use the word leader or politicians instead of the names of people in the title.

The necessity and importance of the study is not properly explained.

Why Philadelphia was studied

What is the importance of the study results?

Who will benefit from the research results?

In the introduction, use more studies and explain the importance of study.

The method part should be described step by step and in more detail.

How was the correct use of the mask by people examined?

Did people know they were being watched by the research team?

People are constantly moving and reorienting, how did you measure the appropriate social distance?

30 locations, why only parks, playgrounds and shopping streets? Are restaurants, passages (shopping malls) and entertainment centers less important?

How can you verify the accuracy of your observations?

How many days were the survey days? How many hours were observed each day?

Were the weather conditions different between the review days and the days before?

Mention study limitations?

Conclusions should be based on the findings of the study.

What are the benefits of the study for the health system?

In the discussion section: in addition to describing the study and its important findings, Compare the findings with other studies and describe the solutions and challenges in this context

Reviewer #3: Title: appropriate

Abstract: appropriate and adequate

Introduction: Authors have indicated the justification to do the study.

Methodology: It was not stated the number of adequate sample size for this research. The justification in choosing the location to be observed was not clearly expelled in the methodology. The characteristics of observers in the study were not clearly stated and various background may promotes bias that may affect the findings of the study. It must be addressed as limitation if there is.

The SOMAD protocol showing that authors made attempt in standardising the research tool and data collection.

Results: appropriate

Discussion: the limitation of the study ie potential bias, limitation on generalisation of the findings were not discussed.

6. PLOS authors have the option to publish the peer review history of their article (what does this mean?). If published, this will include your full peer review and any attached files.

Reviewer #1: **Yes: **Mateus A.

Reviewer #2: No

Reviewer #3: No

---

## [Author Response · Author response to Decision Letter 0]

17 Nov 2021

We revised the cover letter to include the funding statement. 

(RAND is not a commercial company but a non-profit research institute.) 

Reviewer #1: The article presents a relevant and innovative theme. Some small adjustments must be made, they are:

- Improve the introduction, putting an international panorama on the theme (how is the relation of the use of masks in other countries)?

We added Brazil as a country where mask adherence has been politicized. 

- Improve at the end of the discussion, what are the limitations of the study.

We added additional limitations.

- Create a paragraph at the end of the discussion with the practical / clinical implications of your study.

We added this.

- Improve completion by detailing in a topic. Not sure what this means, but hope we provided sufficient detail.

Reviewer #2: I am pleased to share my comment, for the article entitled: Increased Mask Adherence after President Trump Infected with COVID-19

Use the word leader or politicians instead of the names of people in the title. 

We changed to world leader. 

The necessity and importance of the study is not properly explained. 

The importance is due to prevention of spread of a deadly virus. Mask adherence is critical. Countries that have higher adherence have lower case rates. 

Why Philadelphia was studied. 

This was a matter of convenience, it is where our staff was located. We had instituted surveillance prior to President Trumps Covid-19 infection. 

What is the importance of the study results?

The findings demonstrate the importance of messaging and media. People paid more attention when President Trump was infected than to health department warnings. Possibly giving examples and making the consequences more real maybe more effective. 

Who will benefit from the research results?

Leaders, public health professionals and health care providers who want to increase adherence to public health guidance. 

In the introduction, use more studies and explain the importance of study.

We added information on the importance of our study, but there are no similar studies that have used direct observation to monitor mask adherence. 

The method part should be described step by step and in more detail. 

We expanded. 

How was the correct use of the mask by people examined?

By observation. Correct use was defined as covering both mouth and nose.

Did people know they were being watched by the research team? This is unknown. We had no interaction with those being observed. 

People are constantly moving and reorienting, how did you measure the appropriate social distance?

This was a visual estimate.

30 locations, why only parks, playgrounds and shopping streets? Are restaurants, passages (shopping malls) and entertainment centers less important?

We stuck to outdoor locations for safety of the data collectors. 

How can you verify the accuracy of your observations? 

We conducted reliability testing. The results have been published and these are now referenced. 

How many days were the survey days? How many hours were observed each day?

Each site was observed for one hour on the day and time of day over time. 

Were the weather conditions different between the review days and the days before?

Yes, weather follows the seasons and the summer is typically warmer than the fall. 

Mention study limitations? We added some more limitations. 

Conclusions should be based on the findings of the study. We agree

What are the benefits of the study for the health system?

We added a paragraph on the implications. 

In the discussion section: in addition to describing the study and its important findings, Compare the findings with other studies and describe the solutions and challenges in this context.

We are not aware of other published studies that have conducted serial observations of mask adherence. Nevertheless, there are multiple other studies employing direct observation that successfully document behavioral trends.

Reviewer #3: Title: appropriate

Abstract: appropriate and adequate

Introduction: Authors have indicated the justification to do the study.

Methodology: It was not stated the number of adequate sample size for this research. The justification in choosing the location to be observed was not clearly expelled in the methodology. The characteristics of observers in the study were not clearly stated and various background may promotes bias that may affect the findings of the study. It must be addressed as limitation if there is.

Because this is an innovative study there were no prior data informing sample size calculations. Our sample size was based on three considerations. First, the number of observation locations was similar to our previous studies of direct observations of human physical activity behavior in built environment. In many of our past studies, we usually selected anywhere between 6 to 50 sites (e.g., neighborhood parks, recreation centers), in a city for direct observations. Second, the number of observed subjects needs to be sufficient to draw inference for the outcome of interest. Since our outcome is a binary random variable in this paper, a total of 1000 or more subjects yielded sufficient power under the regular power setting of 2-sided p<.05 and power>.8 and for a small to medium effect size. As shown in Table 2, in retrospect we did have sufficient statistical power to declare significance for many substantive predictors. Third, the sample size was also constrained by the available manpower we could deploy during the critical study period. We were not able to further increase the number of locations given the available and trained observers. 

We added this to the limitations. 

The SOMAD protocol showing that authors made attempt in standardising the research tool and data collection.

Results: appropriate

Discussion: the limitation of the study ie potential bias, limitation on generalisation of the findings were not discussed.

We expanded the discussion of limitations.

---

## [Decision Letter · Decision Letter 1]

22 Nov 2021

PONE-D-20-38624R1Increased Mask Adherence after World Leader Infected with COVID-19PLOS ONE

Dear Dr. Cohen,

Thank you for submitting your manuscript to PLOS ONE. After careful consideration, we feel that it has merit but does not fully meet PLOS ONE’s publication criteria as it currently stands. Therefore, we invite you to submit a revised version of the manuscript that addresses the points raised during the review process.

The reviewer has suggested minor revision. Kindly address the comments

We look forward to receiving your revised manuscript.

Kind regards,

Kingston Rajiah

Academic Editor

PLOS ONE

Journal Requirements:

Reviewers' comments:

Reviewer's Responses to Questions

**Comments to the Author**

1. If the authors have adequately addressed your comments raised in a previous round of review and you feel that this manuscript is now acceptable for publication, you may indicate that here to bypass the “Comments to the Author” section, enter your conflict of interest statement in the “Confidential to Editor” section, and submit your "Accept" recommendation.

Reviewer #2: All comments have been addressed

2. Is the manuscript technically sound, and do the data support the conclusions?

Reviewer #2: Yes

3. Has the statistical analysis been performed appropriately and rigorously? 

Reviewer #2: I Don't Know

4. Have the authors made all data underlying the findings in their manuscript fully available?

Reviewer #2: Yes

5. Is the manuscript presented in an intelligible fashion and written in standard English?

Reviewer #2: Yes

6. Review Comments to the Author

Reviewer #2: The title of the study will become more general by changing the form below:

Increased Mask Adherence after Important politicians Infected with COVID-19

Other corrections appear to have been made.

7. PLOS authors have the option to publish the peer review history of their article (what does this mean?). If published, this will include your full peer review and any attached files.

Reviewer #2: No

---

## [Author Response · Author response to Decision Letter 1]

26 Nov 2021

I changed the title as suggested.

---

## [Editor Report · Decision Letter 2]

2 Dec 2021

Increased Mask Adherence after Important Pollitician Infected with COVID-19

PONE-D-20-38624R2

Dear Dr. Cohen,

We’re pleased to inform you that your manuscript has been judged scientifically suitable for publication and will be formally accepted for publication once it meets all outstanding technical requirements.

Kind regards,

Kingston Rajiah

Academic Editor

PLOS ONE
---

## [Editor Report · Acceptance letter]

20 Dec 2021

PONE-D-20-38624R2 

Increased Mask Adherence after Important Politician Infected with COVID-19 

Dear Dr. Cohen:

I'm pleased to inform you that your manuscript has been deemed suitable for publication in PLOS ONE. Congratulations! Your manuscript is now with our production department. 

Kind regards, 

on behalf of

Dr. Kingston Rajiah 

Academic Editor

PLOS ONE